# Inferring experimental procedures from text-based representations of chemical reactions

Alain C. Vaucher [1✉], Philippe Schwaller [1], Joppe Geluykens [1], Vishnu H. Nair[1], Anna Iuliano[2] & Teodoro Laino[1]

The experimental execution of chemical reactions is a context-dependent and time-consuming process, often solved using the experience collected over multiple decades of laboratory work or searching similar, already executed, experimental protocols. Although data-driven schemes, such as retrosynthetic models, are becoming established technologies in synthetic organic chemistry, the conversion of proposed synthetic routes to experimental procedures remains a burden on the shoulder of domain experts. In this work, we present data-driven models for predicting the entire sequence of synthesis steps starting from a textual representation of a chemical equation, for application in batch organic chemistry. We generated a data set of 693,517 chemical equations and associated action sequences by extracting and processing experimental procedure text from patents, using state-of-the-art natural language models. We used the attained data set to train three different models: a nearest-neighbor model based on recently-introduced reaction fingerprints, and two deep-learning sequence-to-sequence models based on the Transformer and BART architectures. An analysis by a trained chemist revealed that the predicted action sequences are adequate for execution without human intervention in more than 50% of the cases.

[1] IBM Research Europe, Rüschlikon, Switzerland. [2] Dipartimento di Chimica e Chimica Industriale, Università di Pisa, Pisa, Italy. ✉email: ava@zurich.ibm.com

In recent years, chemistry has witnessed several successful applications of artificial intelligence (AI) algorithms. Among others, generative models can help design molecules with potentially relevant properties for specific applications[1], while retrosynthetic models suggest potential routes to synthesize these molecules[2,3].

Reaction prediction algorithms assist chemists in prioritizing the synthetic strategy and in selecting effective routes. However, a synthetic route is insufficient to assemble the experimental procedures required for each synthetic step. The planning of a chemical synthesis requires the knowledge of the precise sequence of operations (addition of chemicals, stirring, filtration, solvent extraction, preparation of intermediate solutions, etc.) and the definition of their optimal parameters (temperature, solvents, atmosphere, etc.). The assembly of these operational tasks is left to happenstance, mostly guided by the chemist's experience, and characterized by trial and error. It often requires extensive literature search and the use of homology strategies, in which one identifies one or more reported chemical procedures likely to resemble the target chemical transformation, to provide the best initial guess for the experimental protocol. The reasoning behind this approach is that the execution of a chemical reaction should be successful if one follows procedures for known similar reactions. Additional iteration cycles are often required to improve the reaction protocol after inspection of the experimental results.

Hence, in practice, despite the potential benefits provided by AI algorithms, synthesizing the suggested molecules in the laboratory remains an important bottleneck. This is even more relevant if one considers the revived interest in using robotic systems to automate and scale the execution of chemical reactions[4–12]. In fact, while automation is widely present in chemistry[13], the programming of the robotic systems remains a major impediment towards its wider adoption for general chemistry works due to the need for personnel with both chemical domain expertize and programming skills.

A large-scale adoption of automated synthesis platforms for general purpose chemistry will require virtual assistants to help create specific execution programs for individual chemical reactions. This entails the ability to recommend a precise sequence of operations for the execution of a suggested reaction step, depending on the nature of the substrates, solvents and target products. Concretely, starting from a chemical equation suggested by an AI model, the goal is to determine the series of steps needed to successfully execute that reaction in a laboratory setup.

The use of AI technologies to predict experimental procedures covers, to date, few works focusing on predicting solvents or reaction conditions. For instance, Walker et al. designed a machine learning model to predict adequate solvents for five selected reaction classes[14]. Maser et al. formulated the prediction of reaction conditions as a multiclass prediction problem for four classes of cross-coupling reactions[15]. Their models select categorical values for the categories metal, ligand, base, solvent, additive, temperature, activator, as well as the presence and pressure of a carbon monoxide atmosphere. Nicolaou et al.[8] coupled a retrosynthetic analysis tool to a nearest-neighbor search in a database of previously executed reactions in order to suggest procedures for the automated synthesis of new molecules. Gao et al.[16] trained a neural network to predict reagents, solvents, catalysts, and temperatures for any reaction class. Their model has been combined with retrosynthetic tools to plan syntheses on a robotic platform[5]. However, the chemical procedures had to be revised and complemented manually. The domain complexity and lack of sufficiently curated data hindered further technological developments of AI models predicting entire reaction procedures with limited human intervention.

Here, we present Smiles2Actions, the first AI model to convert chemical equations to fully explicit sequences of experimental actions. We demonstrate it for the realm of batch organic synthesis. The chemical equations, generated by AI algorithms or input by humans, are represented in a text-based format (SMILES). Using a natural language processing model[17], we generate a data set of 693,517 chemical equations and associated action sequences necessary for training three different data-driven models: a nearest-neighbor model, and two transformer-based sequence-to-sequence models, the original architecture as introduced by Vaswani et al.[18] and the bidirectional and auto-regressive transformer (BART) by Lewis et al.[19]. An overview of our approach is illustrated in Fig. 1. When comparing the original and predicted chemical procedures as a whole, the best performing model achieves a normalized Levenshtein similarity of 50% for 68.7% of reactions, a 75% match for 24.7% of reactions, and a 100% match for 3.6% of reactions. The models are able to estimate the solubility of products in different solvents (phase separation, extraction) and to anticipate the formation of

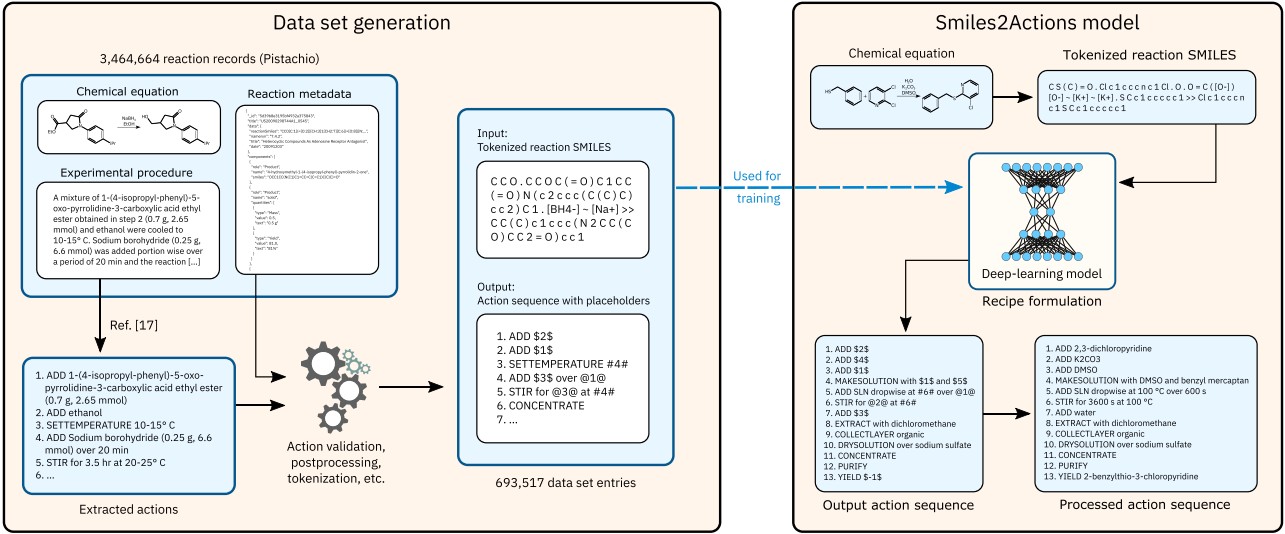

**Fig. 1 Overview of the data set generation and Smiles2Actions model.** The data set is generated in a sequence of processing and filtering steps, starting from information available in patent reaction records (on the left). The Smiles2Actions model is trained on this data set, after which it can predict the action sequences to execute arbitrary chemical equations (on the right).

**Fig. 2 Illustration of a chemical equation.** To the left of the arrow, one can identify all the precursor molecules, while the product molecule is shown to its right. On the left-hand side of the reaction, we also include molecules that play a role as reagents or solvents only, such as the first two entities: N,N'-dicyclohexylcarbodiimide and dichloromethane.

precipitate (filtration), or when to heat or cool the reaction mixture (endothermic or exothermic reactions), without ever making those concepts explicit. Finally, an academic chemist expert analyzes and assesses 500 predicted action sequences among different chemical reaction classes, finding that the predicted action sequences are adequate for execution without human intervention for more than half of the predicted reactions.

## Results

**Prediction task.** We formulate the task of inferring experimental procedure steps as the prediction of an action sequence starting from a chemical equation. The prediction task relates to single reaction steps. For a multi-step synthesis, the experimental procedure steps are predicted separately for each individual reaction.

As in previous works[3,20,21], we do not distinguish between reactants and reagents, as the attribution to one class or the other may be subtle or ambiguous[22]. Hence, the chemical equations in the input consist of a set of precursors (reactants + reagents) and a set of product molecules. Figure 2 shows an example for a condensation reaction.

We depict chemical equations using a text-based representation of the entire set of molecules involved in the corresponding transformation. Without loss of generality, we use the SMILES[23,24] format, which for the example reported in Fig. 2 is equivalent to:

```
C(=NC1CCCCC1)=NC1CCCCC1.ClCCl.CC1(C)CC
(=O)Nc2cc(C(=O)O)ccc21.Nc1ccccc1>>CC1(C)CC
(=O)Nc2cc(C(=O)Nc3ccccc3)ccc21
```

When processing the SMILES representation of a chemical equation, the generated output is a sequence of synthesis actions. The actions follow the format introduced by Vaucher et al.[17] and each consist of a type with associated properties (specific to the action type). A summary of the action types is given in the Supplementary Note 1. These actions cover the most common batch operations for the synthesis of organic molecules, and were designed to contain all the required information to reproduce a chemical reaction in a laboratory. A thorough discussion about the action format is available in the original publication[17].

In order to improve the training performance of the computational models, we restrict the allowed values for two types of properties. The first property relates to the specification of compound names in action sequences. Whenever possible, we use tokens representing the position of the corresponding molecule in the reaction input, allowing the computational models to focus more on relevant instruction patterns instead of trying to learn the naming conventions of molecules. We allow the use of reagents that do not appear in the chemical equation only when they are part of a list of commonly used reagents (reported in the Supplementary Data 1). The second property relates to numerical values for temperatures and durations. The success of reactions does not depend on the exact values for these experimental conditions, as long as they lie within adequate ranges. As a consequence, these values are often reported

inaccurately. An example is the commonly reported reaction duration overnight. The term has little meaning from a quantitative perspective and only indicates the execution of an unattended reaction outside of working hours. The same applies when reporting reaction temperatures. The specification of wide intervals is often a sign of an uncontrolled process, putting aside the systematic errors connected to the use of different measuring devices. Therefore, we tokenized predefined ranges for temperatures and durations and used these tokens during the training process instead of the exact (noisy) reported values. The predicted procedure steps will contain the optimal tokens corresponding to predefined ranges. At inference time, the tokens can be replaced by actual numerical values in a straightforward manner. These two modifications simplify the design and improve the performance of all computational models, as they remove the necessity to learn the vocabulary and syntax of compound names, durations, or temperatures. Also, they reflect the fact that ranges of durations and temperatures are usually as adequate as precise values.

Another important aspect is the mass scale of the chemical transformation. In fact, compound quantities affect chemical procedures and the format designed by Vaucher et al.[17] is general enough to make this relationship explicit. Hence, it may be possible to introduce a functional dependency with respect to mass quantities by specifying an additional reaction token. However, the information extracted from patents lacks a proper coverage across different mass scales to capture the typical patterns of the operational changes when using production quantities compared to laboratory scale[25,26]. Therefore, we removed compound quantities from action sequences, leading to optimal processes that are averaged across the different mass scales.

Taking these points into account, Table 1 shows a possible action sequence for the reaction depicted in Fig. 2.

**Data.** The design of models predicting experimental steps requires a data set of chemical equations and associated experimental procedures. Because of the unavailability of open and large-scale ready-to-use options, we created our own data set of chemical reaction procedures from scratch.

The data set was generated in multiple consecutive steps starting from the Pistachio database[27], which contains 8,377,878 records of reactions published in patents, each including the reaction SMILES string, the experimental procedure, and a mapping of molecular SMILES strings with associated compound names.

Excluding records with no experimental procedure text (2,140,782) and duplicate reaction records (2,772,432), we extracted the action sequences corresponding to the remaining 3,464,664 reactions using a state-of-the-art natural language model (Paragraph2Actions) recently published by our group[17].

The extracted action sequences underwent a series of postprocessing steps to produce a standardized data set of higher quality. We ignored the reaction records reported in languages other than English, records referring to other procedures, and

**Table 1 Possible action sequence for the chemical equation of Fig. 2.**

| | Action sequence | Equivalent human-readable sequence |
|---|---|---|
| 1 | ADD $1$ | ADD N,N'-dicyclohexylcarbodiimide |
| 2 | ADD $4$ | ADD aniline |
| 3 | ADD $2$ | ADD dichloromethane |
| 4 | ADD $3$ | ADD 4,4-dimethyl-1,2,3,4-tetrahydro-2-oxo-7-quinolinecarboxylic acid |
| 5 | STIR for @3@ at #4# | STIR for 8 h at 25 °C |
| 6 | FILTER keep precipitate | FILTER keep precipitate |
| 7 | RECRYSTALLIZE from ethanol | RECRYSTALLIZE from ethanol |
| 8 | YIELD $-1$ | YIELD 4,4-Dimethyl-1,2,3,4-tetrahydro-N-phenyl-2-oxo-7-quinolinecarboxamide |

The tokens $1$, $2$, $3$, and $4$ refer to the compounds present in the chemical equation. Since ethanol is not part of the chemical equation, it is not replaced by a token. The token @3@ refers to the third duration range and corresponds to durations between 3 and 10 h. The token #4# refers to the fourth temperature range and corresponds to temperatures between 10 °C and 40 °C. More details about the token substitution can be found in the Methods section.

**Table 2 Reaction records ignored during the generation of the data set.**

| Category | Number of reaction records |
|---|---|
| Incomplete mapping of molecules | 995,674 |
| Refers to other procedure | 690,484 |
| Contains InvalidAction | 131,461 |
| Error in duration extraction | 127,598 |
| Likely to contain multiple reaction steps | 120,073 |
| Too short action sequence | 67,631 |
| Error in action sequence extraction | 38,516 |
| Error in temperature extraction | 37,485 |
| Molecule present both in the precursors and the products | 6612 |
| Invalid molecule SMILES | 6280 |
| Invalid reaction SMILES | 1606 |
| Other errors | 3544 |
| Removed due to duplicate reaction SMILES | 544,183 |
| Final data set | 693,517 |
| Total | 3,464,664 |

records containing invalid actions upon processing by the Paragraph2Actions model. We used a few heuristics to normalize the action representation and to add implicit actions. We replaced temperatures, durations, and pH values with tokens corresponding to predefined intervals. We matched compound names to molecules present in the corresponding chemical equation, replacing the chemical names with the respective positional tokens. In cases where we were not able to process the chemical records to meet predefined quality standards the entire record was removed from the data set. For records with identical reaction SMILES we retained only one copy. In the Methods section, we report a more detailed description of all the postprocessing steps. The standardization protocol produced a data set consisting of 693,517 reaction SMILES with associated action sequences, a mere 20% of the intermediate set of 3,464,664 records, or 8% of the initial Pistachio database.

In Table 2, we list the leading factors responsible for the decrease of the size of the data set compared to the original number of reaction records. More than one fourth of the initial reaction records is connected to unsuccessful mapping of molecules between the chemical equation and the extracted action sequence. This occurs when a molecule is present in the chemical equation but not in the extracted actions, or vice versa. A similar portion of reaction records was ignored because the corresponding experimental procedures do not contain relevant action sequences upon processing by the Paragraph2Actions model, such as records referring to other procedures, very short action sequences, action sequences with invalid actions, and action sequences that are likely to describe multiple reaction steps. Other errors include flaws in the conversion of extracted duration or temperature strings to actual numerical values.

The last processing step described above, involving the identification/removal of duplicate entries, revealed the presence of 326,929 duplicate reaction SMILES in 871,112 reaction records. Out of these, 47,299 contained non-identical action sequences. The inspection of the corresponding action sequences showed that experimental procedures occasionally describe identical operations using different linguistic terms (for instance, a quenching operation described in terms of addition). The analysis also showed that patent records report identical reactions equally successful under different procedure parameters, such as different temperatures, duration, or the order of the added compounds. While this may be a sign of robustness of some chemical processes when exposed to different conditions, there is no possibility to rule out these records as inaccurate, misleading, or even false reporting. In the Supplementary Data 4, we list all the reaction SMILES and associated action sequences with five or more such sequences. Among the entries with duplicate reaction SMILES, one was picked at random and kept in the data set.

To verify that the data set covers a similar chemical space as the original records, we studied the reaction classes' coverage in the obtained reaction procedure data set. While it is reasonable to observe changes in relative class frequency due to different statistical samplings, it is essential to ensure the absence of systematic bias in our processing protocol. Figure 3 shows the differences in class frequency between the original reaction data with 3,464,664 entries and the final data set of 693,517 reaction records. Most of the reaction classes have a frequency similar to the initial reaction records: out of 944 reaction classes, in 336 cases, the class prevalence changes by <25%, and in 607 cases by <50%. 66 reaction classes, all with a population of fewer than 100 records in the original set, disappeared in the final data set. More than 88.7% of the reaction classes with a size of 100 entries or more in the original data set did not decrease their prevalence by more than 50%. This analysis comprises the class for unrecognized reactions, the frequency of which changes from 30.7% in the original reaction data to 26.2% in the produced data set. Therefore, we consider the generated data set adequately representative of the original reaction data.

In the rest of the work, we use the resulting data set split randomly into 554,813, 69,352, and 69,352 reaction records for training, validation, and testing, respectively. We discuss the class distribution of the different splits in the Supplementary Note 2.

**Models**. We used different architectures to design three computational models for inferring action sequences, given a text-based representation of a chemical equation as input. The first model is

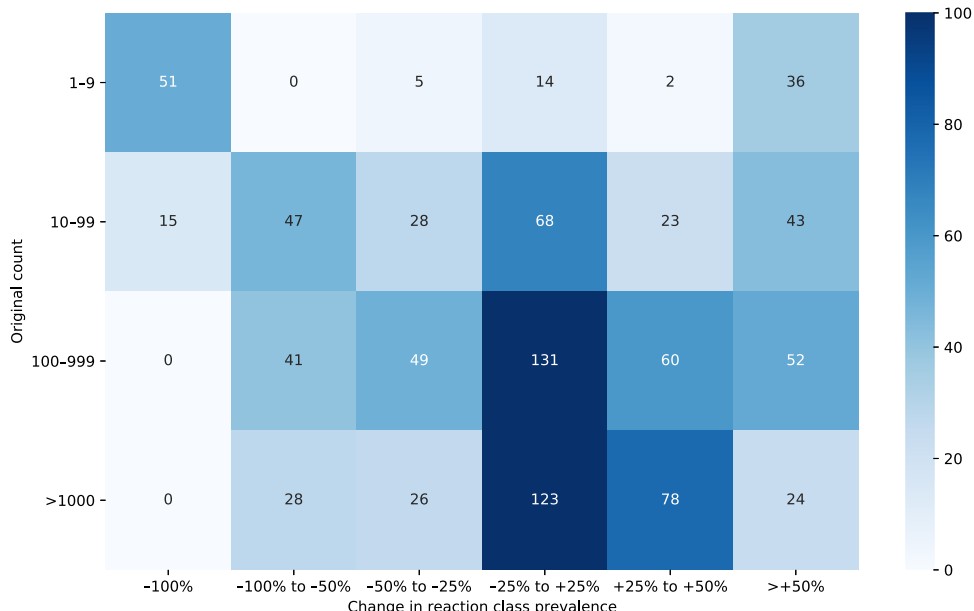

**Fig. 3 Differences in class prevalence between the original data and the generated data set.** As an example, this figure can be read in the following manner: a total of 60 reaction classes, each occurring between 100 and 999 times in the original reaction data set, are represented between 25% and 50% more frequently in the data set of 693,517 reactions.

| Table 3 Metrics for the prediction of synthesis actions. | | | | | | |
|---|---|---|---|---|---|---|
| **Model** | **Validity** | **BLEU score** | **100% accuracy** | **90% accuracy** | **75% accuracy** | **50% accuracy** |
| Random (among all reactions) | 61.6 | 35.1 | 0.00 | 0.04 | 0.76 | 24.07 |
| Random (compatible pattern) | **100.0** | 38.5 | 0.01 | 0.18 | 1.51 | 30.01 |
| Nearest neighbor | 99.6 | 53.2 | **6.65** | **12.50** | 20.30 | 55.46 |
| Transformer | 99.7 | **54.7** | 3.60 | 10.10 | **24.74** | **68.73** |
| BART | 99.6 | 54.5 | 0.98 | 5.00 | 17.57 | 66.04 |

All values are given in percentage, and the best values are indicated in bold. The ground truth is considered to be the only correct solution during the evaluation of the different metrics.

| Table 4 Action sequences predicted for a reaction from the test set. | | | |
|---|---|---|---|
| **Ground truth** | **Transformer model** | **BART model** | **Nearest-neighbor model** |
| ADD $2$ | ADD $2$ | ADD $2$ | ADD $4$ |
| ADD $4$ | ADD $4$ | ADD $4$ | ADD $3$ |
| ADD $3$ | ADD $3$ | ADD $3$ | ADD $5$ at #4# |
| ADD $1$ | ADD $1$ | ADD $1$ | ADD $1$ at #4# |
| ADD $5$ | STIR for @2@ at #4# | STIR for @1@ at #4# | STIR for @1@ at #4# |
| STIR for @4@ at #4# | ADD $5$ | ADD $5$ | ADD $2$ |
| CONCENTRATE | STIR for @4@ at #4# | STIR for @4@ at #4# | STIR for @4@ |
| PURIFY | CONCENTRATE | CONCENTRATE | QUENCH with water |
| YIELD $-1$ | PURIFY | PURIFY | CONCENTRATE |
| – | YIELD $-1$ | YIELD $-1$ | EXTRACT with ethyl acetate/THF |
| – | – | – | WASH with brine |
| – | – | – | DRYSOLUTION over Na2SO4 |
| – | – | – | FILTER keep filtrate |
| – | – | – | CONCENTRATE |
| – | – | – | ADD THF |
| – | – | – | PURIFY |
| – | – | – | YIELD $-1$ |

The considered reaction is a reductive amination of compound $2$ with the amine $4$. The other precursors are acetic acid ($1$), ethanol ($3$), and sodium cyanoborohydride ($5$). The remaining tokens refer to the product ($-1$), to a temperature of 25 °C (#4#), and to durations of 10 min (@1@), 1 h (@2@), and 1 day (@4@).

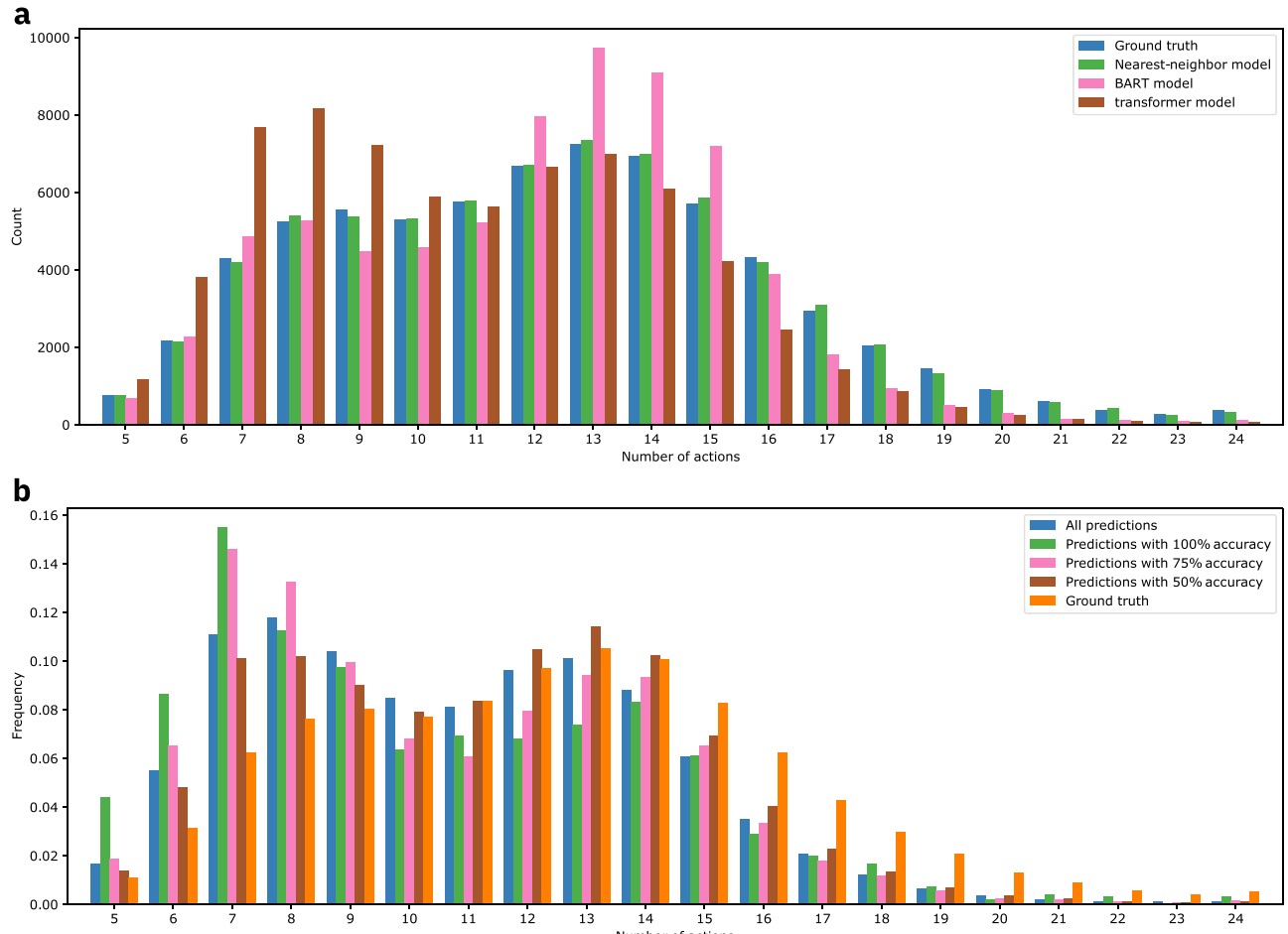

**Fig. 4 Distributions of the lengths of predicted action sequences. a** Comparison of the lengths of predicted action sequences for the different models. **b** Comparison of the lengths of predicted action sequences for different levels of accuracies.

a nearest-neighbor model using novel reaction fingerprints[21] to identify related reactions in the training set and to provide an action sequence adapted from the closest neighbor with a compatible number of molecules in the chemical equation. The second model has a transformer-based encoder-decoder sequence-to-sequence architecture[18] that formulates the procedure prediction task as a translation from reaction SMILES strings to sequences of actions. This model architecture proved successful in earlier work pertaining to SMILES strings[3,20] and synthesis actions[17]. The third model is a bidirectional and auto-regressive transformer model (BART), which builds on the standard transformer architecture and uses recent advances to optimize the pre-training of transformers[19]. The models and their implementation are discussed in the Methods section.

**Model evaluation and comparison.** We evaluate all of the models on the test set containing 69,352 chemical equations and the associated action sequences. In Table 3, we show six metrics for the three models studied in this work. The validity measures the syntactical correctness of the predicted action sequences. It is given by the fraction of predictions that can be converted back to actions (as defined in the Supplementary Note 1) without error, and that contain a reference to all the molecules present in the chemical equation. The BLEU score[28] is an indicator of the similarity between two strings and is a commonly used metric to evaluate models for machine translation. The 100% accuracy refers to the fraction of sentences for which the full action

sequence is identical in the ground truth and in the prediction, including the associated properties. The 90%, 75%, and 50% accuracies are the fractions of sentences that have a normalized Levenshtein similarity of 90%, 75%, 50% or greater, respectively. The Levenshtein similarity is calculated by deducting the Levenshtein distance[29] from one, as implemented in the `text-distance` library[30]. We also include two baselines that randomly pick action sequences, the first one among all reactions from the training set, and the second one among reactions that have the same number of precursors and products.

All three models perform better than the random baseline, which indicates that all of them are able to learn characteristic reaction patterns. The evaluation of the metrics reported in Table 3 requires extreme care as it assumes that the ground truth is the only correct solution to match. However, this is a weak assumption due to possible ground truth errors as well as the multiple possible ways to achieve the desired chemical transformation in the laboratory or to formulate equivalent action sequences. It is a rigid evaluation scheme that penalizes predicted action sequences that may be chemically equivalent but linguistically different. Still, it is the most reliable scheme to provide a statistically meaningful comparison across the different models.

As an example, we report the actions predicted by the different models for a reductive amination reaction in Table 4. The deep-learning models predict a sequence identical to the ground truth except for an additional `Stir` action before the addition of sodium cyanoborohydride. The nearest-neighbor model predicts

**Table 5 Categorization of differences of predictions and ground truth.**

| Category | All properties of other actions are identical | Some properties of other actions are different |
|---|---|---|
| Exact match | 2498 | – |
| Actions in different order | 620 | – |
| Properties of one action are different: Stir | 1935 | – |
| Properties of one action are different: Add | 262 | – |
| Properties of one action are different: Reflux | 163 | – |
| Properties of another action type are different | 234 | – |
| Properties of multiple actions are different | 1123 | – |
| Actions are swapped: Stir and Reflux | 319 | 113 |
| Actions are swapped: Stir and Microwave | 99 | 11 |
| Actions are swapped: Stir and Wait | 70 | 64 |
| Actions are swapped: Add and MakeSolution | 118 | 277 |
| Other swap of a single action | 442 | 2207 |
| Action without counterpart: Purify | 286 | 747 |
| Action without counterpart: Wash | 154 | 737 |
| Action without counterpart: Set Temperature | 175 | 579 |
| Action without counterpart: Filter | 121 | 462 |
| Action without counterpart: Concentrate | 173 | 404 |
| Action without counterpart: Stir | 154 | 313 |
| Action without counterpart: Add | 117 | 308 |
| Action without counterpart: Collect Layer | 48 | 207 |
| Another action type without counterpart | 191 | 550 |
| Multiple actions only in the ground truth | 2027 | 8475 |
| Multiple actions only in the prediction | 401 | 2161 |
| Remaining cases | 7280 | 32,727 |

The differences are computed for the test set containing 69,352 reaction records.

a longer action sequence including quenching and a more involved work-up. It is noteworthy that all the models predict an identical stirring duration. In the Supplementary Data 2, we report a selection of 100 reactions and associated predictions, including the one from Table 4.

Figure 4a shows the distribution of the number of actions predicted by the different (non-random) models. The sequence lengths of the nearest-neighbor model closely follow the distribution of the ground truth. The transformer model has a pronounced preference for shorter sequences, while the BART model is biased towards middle-length sequences. Both transformer and BART models predict fewer sequences that include 15 or more actions.

Figure 4b shows the distribution of the lengths of action sequences predicted with accuracies of 100%, 75% and 50% using the transformer model. The correct (100%) predictions cover a similar range of sequence lengths as the full data set and are not limited to short sequences. Short sequences are only slightly overrepresented, which is consistent with the higher probability of predicting short sequences correctly.

The fact that the proportion of chemical equations for which the prediction matches the ground truth 100% is under 10% for all models does not mean that their quality is poor. The low percentage of exact matches can be explained, in part, by the presence of sequences with a large number of steps negatively affecting the percentage of entirely correct sequences (multiplicative reduction of the probability if the events were independent), by the noise in the underlying data set (multiple correct ways of doing a reaction, multiple ways of describing the same action sequence) and by errors in the data set (errors in the action sequence extraction from the experimental procedure text). For similar reasons, the differences in the distribution of action sequence length in Fig. 4a are not a valid argument to judge the quality of the deep-learning models compared to the nearest-neighbor scheme.

In an endeavor to rationalize the model predictions for single actions, one can assume, for illustration purposes, that the prediction of each action is an independent event. We used the average value of the accuracy for the 100% match and the distribution of the sequences' length to calculate the accuracy of a single action prediction by solving the polynomial equation that sums the probabilities of exactly matching each sequence length to a total value of 3.6%. We found an accuracy of 72.7% for the average single action prediction, which would correspond to the average probability of the model predicting a single action with correct type and associated properties if the action predictions were independent events.

Despite the intrinsic difficulties to assess and compare the absolute performance of the models, it appears that all three schemes are viable options for inferring action sequences from chemical equations. In general, the transformer model performs slightly better than the BART model. The nearest-neighbor model has a better 100% accuracy value (see Table 3) than the deep-learning models. The deep-learning models, however, rely on a learned representation rather than on similarities with data points in the training set. As such, they automatically take different aspects of the input into account, such as the type of transformation and presence or absence of functional groups. This makes them more general and more interesting in the context of experimental procedure prediction. For this reason, we will consider only the transformer-based model for further analysis.

In Table 5, we categorize the differences in the actions predicted by the transformer model compared to the ground truth. In addition to the 3.6% exact predictions, roughly 0.9% represent actions predicted in different orders, with the Add action being the main entity affected by incorrect orderings. In 5.4% of the action sequences, the differences are limited to properties associated with the action types. For 1.4% of reactions, one action is predicted instead of another one - and for 3.9% other cases, some properties of other actions are different in addition to the swap. The swap between similar entities

| Table 6 Result of the chemist's assessment of action sequences. | |
|---|---|
| **Decision** | **Number of reactions** |
| Both sequences are adequate | 191 |
| Predicted action sequence is adequate, ground truth is inadequate | 122 |
| Predicted action sequence is inadequate, ground truth is adequate | 108 |
| Both sequences are inadequate | 79 |

Out of the 500 analyzed reactions, 19 had an identical action sequence in the ground truth and in the prediction, 16 of them were considered adequate and 3 were considered inadequate.

(such as Stir and Reflux) may indicate the minor importance of the use of a specific action type for a reaction success. In 8.3% of the reactions, an action is missing either in the ground truth or in the predictions. Most often, the extra action is related to work-up or purification. In 18.8% of cases, more than one action is missing in either the ground truth or in the predictions. The remaining 57.0% of the reactions show a combination of multiple types of differences.

For a set of 500 reactions, we did a blind human assessment of the action sequences reported in the ground truth and predicted by the transformer model. For each reaction, we showed a trained organic chemist a chemical reaction diagram together with the ground truth and the predicted action sequences, in a random order. The domain expert, without any possibility to distinguish the predicted sequence from the ground truth one, was asked to judge whether the experimental procedures were adequate or not by selecting among the following options: consider both of them experimentally valid, consider one to be adequate and the other one inadequate, or reject both as experimentally inadequate. Often, the expert labeled sequences as inadequate even if only a single action was considered to be incorrect. We report the collected preferences in Table 6. While this test cannot be considered statistically significant to assess the quality across different chemistry areas, it is unquestionable that for an academic level expert the predicted procedures are, on average, of equivalent quality to the ground truth. For more than half of the reactions, the chemist considered the predicted action sequence to be adequate. For about two fifths of the reactions, the chemist considered them inadequate. Moreover, the human assessment indicates that the predictions contain slightly fewer inadequate sequences than the ground truth. Therefore, it is reasonable to assume that improving the quality of the ground truth will result in an improvement of the model predictions. We report the human assessment and corresponding analysis of the 500 reaction records in the Supplementary Data 3.

The Supplementary Information contains additional evaluation information for the transformer model. In the Supplementary Note 3, we report the effect of shuffling the molecules in the reaction SMILES given as input. We illustrate the performance of the model with respect to the reaction class in the Supplementary Note 4, and present the class distribution of the reactions assessed by the expert chemist in the Supplementary Note 5.

## Discussion
In this work, we address the prediction of experimental steps starting from a text-based representation of a chemical equation. We generated a data set beginning with a database of patent reactions to train three predictive models: a nearest-neighbor scheme, and two deep-learning sequence-to-sequence models based on the Transformer and BART architectures. Despite the mathematical differences, the three models exhibit a similar performance.

For most reactions in a random selection of the data set, the predicted action sequences are considered experimentally adequate (313 out of 500) by an expert chemist in a blind assessment. The remaining 187 predictions out of 500 are considered inadequate. The same assessment reveals the presence of 201 inadequate procedures in the ground truth. We demonstrated that it is possible to construct effective deep-learning schemes to learn the characteristic patterns in chemical reaction procedures and, at the same time, we showed that quality issues within the information extracted from patents limit their potential. An inspection of the inadequate procedures in the ground truth highlights some shortcomings in the underlying patent data, such as incorrect chemical equations, but also hint at improvable weaknesses in the generation of the data set. A reduction of the number of inadequate data procedures could for instance be achieved by an improved detection of inexact reaction data, and by further improving the action extraction procedure from experimental reports. Seeing that the transformer model is performing on par with the ground truth, we expect that the model performance will strongly benefit from such quality improvements in training data.

Without loss of generality, we simplified a few aspects of the reaction procedures. We did not include any information about the state or concentration of compounds, as this is not commonly specified in chemical equations. However, the current models could easily be broadened with the availability of higher quality data sets and an extension of the current text-based representation to include the compound state or concentration. The same is true for the relative quantities of precursors and solvents, characterizing the reaction scale, and for the atmosphere under which reactions are executed. These pieces of information are not reported uniformly in the experimental procedures and their use would degrade the overall performance of deep-learning schemes. Therefore, we decided to remove this information during the data standardization process. This decision led to some incomplete action sequences, especially for hydrogenation reactions that take place under hydrogen atmosphere.

We note that action sequence prediction models require chemical equations to include not only reactants and products, but also solvents and reagents. Recently developed retrosynthetic schemes[3] are capable of predicting optimal solvents and catalysts and thus automatically provide the chemical equation in the required format. A recently reported algorithm can be used to infer common reagents and solvents automatically if they are not present in the chemical equation[31].

Interestingly, a chemical procedure is the equivalent of a computer program for experimental chemists: a series of instructions specified in a human-readable format that unambiguously codifies the operations to execute the chemical experiment[17], which could either be executed by human operators or by automation hardware. Accordingly, the Smiles2Actions model can be considered to write code for chemical synthesis, and therefore has similarities with machine programming[32], where the core idea is software creating its own software. The use of artificial intelligence technologies for inferring experimental procedures will reduce the amount of trial and error in a traditional laboratory setup. When coupled with an automation system, this technology will contribute to a wider adoption of automation technologies, laying the foundations for a fully automated synthesis starting from only chemical equations. In fact, we strongly believe that the mathematical architectures presented in this paper will become an essential component to automatize general purpose synthetic chemistry on robotic systems. AI will not replace chemists, and the action sequences predicted by the model introduced in this work should always be verified for safety prior to the actual synthesis. But AI will soon reach a level where the predicted experimental procedures will be production-worthy, without requiring human intervention, and will directly be usable to drive automation hardware in a chemical laboratory or to reduce the amount of trial and error in a traditional laboratory setup.

## Methods

**Action sequence extraction.** We used the pristine natural language processing model trained in ref. [17] (Paragraph2Actions) for the extraction of action sequences from experimental paragraphs.

**Handling of compound names.** The identification of the compounds present in the action sequences extracted from experimental procedures requires the mapping of the compound names to chemical structures. In the next paragraphs, we document the detailed procedure.

*Compound name stripping.* In Vaucher et al.[17], the action sequence extraction model was trained to extract explicit compound names. For instance, the sentence "The solution was acidified with 12 mL of a saturated solution of sulfuric acid" contains the extracted compound name "saturated solution of sulfuric acid". The mapping of extracted names to SMILES strings includes the identification of the root compound name, in this case "sulfuric acid". Also, in several cases, the extracted compound names refer to multiple chemical compounds, as in "5 mL of a 1.0 M DCM solution of boron tribromide" or "80 mL of 1:1 dioxane-water".

To obtain the root compound names, we built a series of data-cleaning steps based on heuristics (https://git.io/JYWDY) to trim the compound name entities by removing, among others, information about physical state ("solid", "gaseous", "(s)"), concentrations ("1.0 M", "10 wt%", "saturated", "concentrated"), temperatures ("cold", "hot"), or composition ratios ("1:1", "2/1", "(80:10:10)").

*Compound name normalization.* Chemical compounds often have multiple names. In addition to common synonyms (for instance: "isopropanol", "2-propanol", and "propan-2-ol"), the differences among extracted entities are commonly limited to single characters, often as a side-effect of the use of optical character recognition tools in scanned documents. Normalizing compound names with respect to such small differences simplifies the attribution of SMILES strings to compound names.

We implemented a pipeline (https://git.io/JYWMt) for the normalization of compound names, in which they undergo several subsequent modifications:

- subscript digits are replaced by their ASCII equivalent: $H_2SO_4 \rightarrow H2SO4$.
- visually similar characters are unified to avoid common optical character recognition pitfalls: I (uppercase i) → l (lowercase L), 1 (one) → l (lowercase L), 0 (zero) → O (uppercase o).
- uppercase characters are converted to lower case.
- dash and prime characters are unified or removed.
- spelled-out Greek letters are replaced by the Greek letter: alpha-glucose → α-glucose.
- spaces are removed: dimethyl sulfoxide → dimethylsulfoxide
- a selection of special characters are removed: $NiCl_2 \cdot 6H_2O \rightarrow NiCl26H2O$, $NiCl_2 \times 6H_2O \rightarrow NiCl26H2O$

Note that the resulting names may not be existing compound names. However, the purpose of the compound name normalization is to have a common representation language for names that may differ due to spelling errors.

*Name-to-SMILES and SMILES-to-name.* In order to match the molecules between the text-based representation of the chemical reaction and the extracted experimental procedure, we implemented a mapping scheme between compound names and SMILES strings.

We built two mappings using pairs of compound names and SMILES strings available in the Pistachio database with hash tables (Python dictionaries). They contain the most common SMILES string for a given normalized compound name, as well as the most common compound name for a given SMILES string. We called the two services name-to-SMILES (https://git.io/JYWMR) and SMILES-to-name (https://git.io/JYWMo), respectively.

By mapping a compound name to a SMILES string and then back to its compound name, one can obtain the most common synonym for the given compound name. For instance, "dimethyl sulfoxide" is mapped to the SMILES string "CS(C)=O", which maps back to "DMSO".

**Tokenization of temperatures, durations, and pH values.** We replaced the extracted temperatures, durations, and pH values with tokens representing predefined intervals (https://git.io/JYWMS). We report, in the Supplementary Note 6, the ranges, tokens, and values used to convert the predicted instructions into numerical values.

Prior to the actual tokenization, we needed to convert the strings of characters representing temperatures, durations, and pH values to actual numbers with associated dimensions. This was achieved with heuristics that account for the multiple ways of representing numbers ("five", "5") and units ("C", "F", "degrees", etc.), and that assign predefined values when no actual number is given ("over the weekend", "overnight", "RT", "ambient temperature", etc.). If the conversion of an extracted value failed, we ignored the associated reaction and did not include it in the data set of experimental procedures.

**Data set generation.** As a source of chemical reaction data, we selected the Pistachio database, version 3.0[27]. Duplicate reaction records and records with no experimental procedure text were filtered out. This provided a set of 3,464,664 reaction records.

The application of the following standardization steps produced 693,517 reaction SMILES and associated action sequences to train the machine learning models.

**Reaction SMILES postprocessing.** The Pistachio database provides reaction SMILES strings parsed into reactant, reagent, and product molecules. We merged the reactant and reagent molecules into a list of precursor molecules, and all the SMILES strings were canonicalized with RDKit[33]. For both lists of precursor and product SMILES, we removed the duplicates and reordered the lists alphabetically. The concatenation of the SMILES strings produced the reaction SMILES used for training. Following the reaction SMILES notation, we separated the molecules within the same class using dots ("."), while the precursor and product lists were separated by ">>". For fragment bonds, we adopted the convention of using the tilde symbol ("~") instead of a dot.

For use in language-based models, the reaction SMILES is tokenized by inserting spaces between the SMILES tokens.

**Action sequence filtering.** Some action sequences are not adequate for training and were removed (https://git.io/JYWDi) from the data set. This is the case for reactions with incomplete experimental procedures (such as the ones shortening the description by referring to other procedures), or for unsuccessful or incomplete action sequence extraction.

We ignored experimental procedures when their processing with the Paragraph2Actions model[17] contained any InvalidAction or FollowOtherProcedure action. We filtered out any experimental procedure containing fewer than five actions because such short sequences are not likely to describe a chemical reaction appropriately. We also ignored experimental procedures that likely describe multiple reaction steps. Therefore, any experimental paragraph whose action sequence contains multiple Yield actions interlaid with actions other than purification or work-up were rejected.

**Action sequence postprocessing.** The action sequence extraction model operates sentence by sentence. When combining the actions for a full experimental procedure, it is crucial to provide a consistent record, guaranteeing correct relative dependencies between instructions. To do so, the action sequence extracted from the experimental procedure text underwent a series of postprocessing steps (https://git.io/JYWyd).

We removed all NoAction actions, because these instructions usually arise from sentences that are not relevant to the actual synthesis and can safely be ignored.

Whenever possible, we merged Wait actions with previous actions. This provides a more concise representation when a sentence specifies the duration of an action only in the next sentence. For instance, "The mixture was brought to reflux. After 2 hours, ..." is equivalent to a Reflux action with a time parameter of 2 h.

Often, experimental procedures do not explicitly state whether the precipitate or the filtrate should be kept upon filtering. We therefore inferred this information from the previous or subsequent actions.

The use of preceding actions is also important to complete instructions containing information related to previous actions. For instance, an action specifying the temperature parameter with the "same temperature" value, requires the inspection of the preceding actions to identify the latest set temperature.

Finally, we replaced MakeSolution actions appearing as first instruction in a procedure by a series of Add actions. This reduces the differences between equivalent ways of formulating action sequences while keeping an identical logic in terms of experimental operations.

**Updates to individual actions.** To further remove unnecessary pieces of information and harmonize differences in action sequences, we applied a series of changes to the individual actions (https://git.io/JYWyd). First, we removed all compound quantities. The inclusion of mass information would only decrease the overall accuracy of the model. Better and more evenly distributed data sets will make it possible to skip this simplification step, thereby enriching future predictive capabilities. Second, we tokenized all the values for temperature, duration, and pH values as described above. Third, because of the uneven reporting of the number of repetitions for Extract and Wash actions, we decided to ignore these values and always assume a single repetition. Finally, again because of poor reporting consistency of experimental details, we ignored all indications related to the use of specific atmospheres except when the reported value was "vacuum" for the DrySolid action.

**Compound names substitutions.** As explained in the Results section, compound names are only admissible in action sequences if the corresponding tokens refer to molecules in the chemical equation or to molecules in a list of common reagents. Therefore, it is necessary to map the molecules present in the chemical equation to the extracted compound names. If the mapping is unsuccessful, we verify if their most common synonyms match any item on the list of common reagents. We report the list of common reagents in the Supplementary Data 1. The generation of this list is explained in the Supplementary Note 7.

In addition to the name-to-SMILES and SMILES-to-name mappings introduced above, the Pistachio records provide reaction-specific mappings between compound names and SMILES strings. These mappings are particularly useful when compounds are specified with a general name such as "the compound from Example A", "title compound", etc. The mappings from Pistachio have precedence over name-to-SMILES and SMILES-to-name.

We trimmed each compound name in an extracted action sequence and mapped its normalized name to a SMILES string, if possible. We separated extracted compound names containing more than one compound by a slash ("/") symbol. If the SMILES string corresponded to a molecule present in the chemical equation, we used a placeholder indicating the position in the chemical equation instead of the extracted compound name. If the SMILES string did not correspond to any molecule in the chemical equation, we converted it to its most common synonym; for instance, "dimethyl sulfoxide" is replaced by "DMSO".

We only kept compound names that did not match any molecule in the chemical equation if they were present in the list of common reagents.

Upon application of these guidelines, if exactly one molecule from the chemical equation and exactly one extracted compound name had not been mapped yet, we considered them to be matching entities. We ignored the entire experimental procedure if, at the end of this protocol, there were still unmapped molecules or extracted compound names.

*Textual representation of action sequences.* As in previous work[17], we converted the actions (types and associated properties) to a semi-structured textual representation. This format is concise, human-readable, easily understandable and it enables the use of natural language machine-learning models. This representation can be converted to and from the action type and associated properties without loss (https://git.io/JYW9e). The actions listed in Tables 1 and 4 use this textual format.

## Models

*Nearest-neighbor model.* The nearest-neighbor model relies on the computation of reaction fingerprints (`rxnfp ft`), as published in our previous work[21], for the training and test set. For any test set reaction fingerprint query, we search the nearest neighbors in the training set with `faiss`[34]. The nearest-neighbor search is constrained to the training reactions that have the same number of precursors as the test reactions.

The data-driven reaction fingerprints from Schwaller et al.[21] were preferred to the connectivity-based reaction fingerprints from Schneider et al.[35]. The latter require a distinction between reactants and reagents, which may be subtle or ambiguous[22], and have a lower accuracy than the data-driven fingerprints on the reaction classification task[21].

*Transformer model.* The transformer model uses a transformer encoder-decoder architecture with 8 attention heads and is trained by minimizing the categorical cross-entropy loss for the output words. The model was implemented with the `OpenNMT-py` library[36,37]. We adopted the hyperparameters suggested by the library with a few changes. First, we reduced the model size by decreasing the number of layers from 6 to 4, the size of the hidden states from 512 to 256, and the size of the word vectors from 512 to 256. Second, we changed the values of the parameters `max_generator_batches` to 32, `accum_count` to 4 and `label_smoothing` to 0.

*BART model.* The BART model was implemented using the reference implementation in the `fairseq` framework by Facebook AI Research[38].

## Data availability

The generated data set of 693,517 chemical equations and associated action sequences is a derivative work of the Pistachio data set[27]. It is available from the authors upon request.

## Code availability

A GitHub repository, available at https://github.com/rxn4chemistry/smiles2actions[39], contains code for processing compound names and action sequences, as well as instructions to train and use the transformer model presented in this manuscript.

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

## Acknowledgements

We thank the anonymous reviewers for their careful reading of our manuscript and their many insightful comments and suggestions.

## Author contributions

The project was conceived and planned by T.L. and A.C.V. and supervised by T.L. A.C.V. generated the dataset and analyzed the results. A.C.V. and V.H.N. trained the transformer model. P.S and J.G. implemented and trained the nearest-neighbor and BART models, respectively. A.I. assessed the subset of 500 reactions and action sequences. All the authors were involved in discussions about the project. A.C.V. wrote the manuscript with input from all authors.

## Competing interests

The authors are listed as inventors of a filed patent by IBM based on this work (application number 16995853).
