## [Peer Review File · Nature Communications]

REVIEWER COMMENTS

Reviewer #1 (Remarks to the Author):

The manuscript describe an interesting model for prediction experimental action sequence from a chemical equation. Although the model quality still have much room to improve, this is an important step toward translation of chemical reaction to machine readable/actionable languages. I think the idea is very novel, the manuscript is well written and in general meet the quality for publishing in Nature Communication. I have following comments for the authors to consider:

1. I am wondering if the authors can check if the order of reagents and reactants in the input equation can affect the model prediction.
2. The model performance according to the reaction class should be discussed. How many reaction classed included in the training set and test set? What is the reaction class overlap among training and test set?
3. I am wondering if a transformer pretrained on chemical text can improve the prediction, can the authors comment on that?

Reviewer #2 (Remarks to the Author):

Vaucher et al. showed that natural language processing techniques can predict the sequence of reaction steps, given reactants and products in SMILES representation. They showed that a trained chemist believes that the machine-predicted synthesis actions are ready for execution in a majority of cases.

I am afraid this paper is significantly below the threshold in terms of significance and impact. As such, I regrettably recommend rejection.

Reaction condition prediction is not a new problem. Walker et al. and Gao et al., cited as ref 5 and 6, already showed that aspects of reaction condition can be predicted. The authors argue that the prediction is not done to the same level of granularity, which is true. However, the authors need to show that this extra level of granularity is accurate, thus meaningful.

I think the authors included the expert judgement test to demonstrate chemical accuracy. However, I'm afraid a human chemist would not be able to eyeball the exact sequence of steps. In fact, this is borne out by their data. Their chemist concluded that in 79/500 cases, both the "ground truth" and the predicted sequences are inadequate, despite one of the sequences is in fact from the literature. Therefore, the human test is weak, and insufficient to justify that the extra level of granularity is actually accurate.

I would suggest that the meaningful test is (perhaps unfortunately) prospectively doing the experiments. This was the approach taken by several recent papers, e.g. Coley et al. (ref 7), and in the context of retrosynthesis, Mikulak-Klucznik et al. (<https://doi.org/10.1038/s41586-020-2855-y>).

If the authors are planning to do experiments, I suggest:

(a) Trying test set reactions where the model predicts a different sequence of steps compared to literature, and

(b) Trying reactions on unseen substrates, e.g. 4-5 mechanistically different but common reaction classes, and 4-5 reactions per class, making sure that the substrates are chemically very different from the any substrates in the dataset. (b) is important because both the training and testing reactions are "successful" reactions (at least good enough to be mentioned in a patent). This is a significant bias, because reactions often fail!

Reviewer #3 (Remarks to the Author):

The authors have designed a method that can create recipes for executing chemical synthesis. The paper is well written and certainly publishable in a more technical journal, however, it seems to lack the novelty needed to be publishable in Nature Communications.

Arguably the most interesting part, the text based extraction of the reaction data, has already been published in Nature Communications, reference [9] in this manuscript.

It is truly remarkable that the the recent article from the Cronin lab is not cited
<https://science.sciencemag.org/content/370/6512/101/tab-article-info>

The article has very similar goal as this one and in addition includes experimental validation. The article must be known to the authors since one of them has been engaged in a discussion with Professor Cronin on twitter.

Reviewer #4 (Remarks to the Author):

The manuscript titled "Inferring experimental procedures from text-based representations of chemical reactions" builds on previous work by the same authors by extending extracted chemical procedures of exact molecules to a machine learning model that is able to predict procedure operations for unreported molecules. They compare their transformer model to a baseline that mimics how chemists generally apply known procedures to new compounds using a nearest neighbor search based on chemical structure. The analysis for chemical operations can be very difficult since in many cases, there are a multitude of different procedures in various different orders that may be successful for the desired chemical transformation. The authors do a thorough analysis despite the difficulty in analysis.

The article is suggested to be published after addressing the following concerns.

Overall:

The publications on experimental procedures tend to be getting further and further away from being accessible or even usable for the chemistry community. The following make it difficult to implement and validate results:

- 1) The procedures sequences seem to be tailored for an IBM type automated system. More specifically, only batch chemistry with an ability to program a robot to do these specific actions. This can be acknowledged in the introduction when the authors state "the programming of the robotic systems remains...". This is important because end users need to know the domain of applicable systems where the predictions can be leveraged. For example, these actions could not be used for flow, parallel chemistry, or HTE in the current form (one could envision how to extend to other systems, but not possible with these action sequences).
 - 2) Use of closed source data. The reviewer understands that there are not other high quality sources of procedure data available but
 - 3) Use of NextMove software to resolve chemical names. While seems robust this is just another closed source package a user would need to have access to in order to repeat this work with their own data. Are there discussions with NextMove to open source the mappings? If not, this could be changed to using the open source PubChem API for resolving synonyms. Eg. For iPrOH
<https://pubchem.ncbi.nlm.nih.gov/rest/pug/compound/name/iPrOH/synonyms/JSON> then you can resolve smiles by using the CID
<https://pubchem.ncbi.nlm.nih.gov/rest/pug/compound/CID/3776/property/CanonicalSmiles/JSON> where you can replace the number after CID/ to get the smiles for the compounds. This at least would give the ability for others to use the software, even if they can't use the same dataset.
- From the chemist's viewpoint, accuracy for procedures is very important. Depending on the reaction type, safety is a really big issue. It is understood that the ML models should learn compatibilities such as a reagent that decomposes at high temperature would never be recorded so the model should not propose it. However, from a risk assessment view in a chemistry department or in industry, there needs to be VERY compelling evidence that these incompatibilities are never executed even robotically where an explosion or fire could hurt human operators and damage equipment/buildings.

Introduction:

- "The reasoning behind this approach is that the execution of unknown chemical reactions should...". In this sentence "unknown chemical reactions" should be changed. While the reviewer understands what is being stated, a chemist views this as an undiscovered reaction (ie new reaction type or class) when the authors are really trying to say novel reactant combinations but still using known reaction types.
- "Hence, in practice, despite the considerable benefits provided by current AI algorithms...". Most readers will disagree since there has not been a large demonstrated impact (yet) of AI on synthesis planning and execution. Mostly what we see is vignettes of one-off cases where AI helped. Citations would be needed

to backup this claim OR change the sentence to “despite the possible benefits provided by current AI algorithms”.

- The comparison of ML procedures to machine programming is not complete. The similarities are there but it is not clear what the authors are trying to state with this comparison. If it is just to acknowledge that they are building on previous knowledge in other fields, then that is understandable. However, chemists might not see the connection and the major difference in the two predictions is that one misprediction in chemistry can result in major safety issues where in programming there is lower risk associated with poor predictions.

- “The models do not contain any human encoded chemical knowledge...” While this is true, everything still relies on human coded SMILES (which is a syntax the model is learning), human coded extraction of data etc... This is not to say that this should necessarily be changed but to alert the authors that there are other interpretations.

- Distribution of reactions that the expert chemist looks at.

Results:

- How is the “commonly used reagents” determined? Frequency? If so, what is the threshold?

- There is a lot of discussion in the throughout the whole manuscript about the inaccuracy of “overnight”. To a chemist this actually gives a wealth of information, even if it is not useful for automated extraction of procedures. Information gathered from overnight could be that duration is probably between 8-16 hours, an 8 hour range which is not that different of a range than the authors @3@ token (7 hours) and is more specific than the @4@ token. Chemists also gather that the reaction is very likely safe for extended periods of time without supervision and the reactants and products do not decompose over that time. Despite the limitations of overnight in text, the authors defined ranges do not seem to be much more specific and hence the long discussion unwarranted. I think the authors are just trying to state the difficulty in extraction since procedures are very heterogeneous and not standardized.

- “These two modifications simplify the design and improve the performance of all computational models without a significant impact on the quality of the predictions.” Please define what “significant impact” is. How is this measured? Quantitatively or qualitatively?

- “However, the information extracted from patents lacks a proper coverage across different mass scales to capture the typical patterns of the operational changes when using production quantities compared to laboratory scale” cite relevant literature on dataset problems. There are plenty of perspectives and reviews that also call this out. Or cite relevant literature about dataset comparisons in the chemistry realm.

- Are the data splits stratified with respect to the data classes or random? If random, provide the resulting data distributions for train, valid, test.

- Figure 2: does this analysis include reaction class 0.0 unrecognized?

- It is nice to use the novel fingerprints from ref 13 but it would be advisable to use traditional fingerprints (eg circular/Morgan from RDKit) as well. These would give results that would likely be more similar to how search engines such as Reaxys or SciFinder work which allows for those familiar with this type of nearest neighbor search to better interpret the results.

Model evaluation and comparison:

- “Assuming the prediction of each action is an independent event...” Frankly, this is an erroneous assumption. The whole reason to use a transformer model for sequence translation is because each prediction relies on previous and subsequent predictions, hence the global attention heads. It is the reviewer opinion that this analysis does not add much significance and should be removed which would not impact the manuscript as a whole.

- “However, deep-learning models are preferable because they rely on a learned representation rather than on similarities with data points in the training set.” This is debatable. As stated above from a safety risk assessment point of view, most likely close to 100% of companies would go with nearest neighbor model because there would be less liability.

- Figure 4 add the ground truth to the graph

- Chemist analysis. This is a very statistically unsound analysis. A single chemist cannot be the judge of the action sequences. The only result the authors should even claim here is if the sequences are adequate or not. The ranking of which prediction is better should be removed OR the authors should recruit 5 or more chemists, demonstrate the chemists have no ties with their research and thus should be less biased, and get their rankings (and even 5 is probably a pretty low number to draw a conclusion on). Chemists have natural biases on how they run reactions which is clearly evidenced by the multiple recorded sequences for the same reaction in the database.

Discussion:

- "For most reactions in a random selection of the data set, the predicted action sequences are considered experimentally adequate (313 out of 500) by an expert chemist in a blind assessment. The remaining 187 predictions out of 500 are considered inadequate. The same assessment reveals the presence of 201 inadequate procedures in the ground truth" If the authors are to claim these 500 predictions are representative, then the natural thought to the reader is that 201 out of 500 for the ground truth should be representative as well. This is 40% of the dataset that is not fit for use. This calls into question the methods that are used to extract the sequences. The authors should comment about this in greater detail.

Methods:

- "The mapping of extracted names to SMILES strings includes the identification of the base compound name, in this case "sulfuric acid"." Replace the word base because when acid is mentioned one thinks of chemical base. Maybe "root compound name".

- It would be nice to see how many different ways concentrations, composition ratios etc. are reported. This would be of great interest to the data scientists and should be fairly trivial to report the number of different ways these are reported, or at least how many the authors consider for standardizing.

- Multiple sequence entries for the same reaction:

1) What is the criteria for keeping a sequence if the reaction smiles has multiple different entries? Shortest sequence?

2) A weight or score could be applied to reactions that have many entries. Could you add a "robustness" score that is trained alongside of the transformer model? Have some range determined by the min and max number of recorded successful sequences for unique reaction smiles. The output would be sequences and how they would score on a robustness scale. This would be very useful to chemists because it mimics how they would look for procedures to use. For example, they look up an amide coupling and using the same precursors (reagents + reactants) there are 50 reported procedures all that differ. They can infer that it is a robust transformation and will be fairly insensitive to order of addition. You would also be able to use the score to investigate "incorrect" predictions. Hopefully the authors would find that incorrect predictions with a high robustness score would only differ by a trivial order of addition (where safety would be of no concern) or another simple change from the ground truth.

References:

The references are a little sparse. There is a plethora of work in automating chemistry and since that is the end goal of this work, it should be acknowledged. Here are a few to start but there are many references contained in these that should be acknowledged as well.

Lilly automated planning and execution:

<https://pubs.acs.org/doi/full/10.1021/acsmchemlett.8b00488>

<https://pubs.acs.org/doi/full/10.1021/acs.jcim.9b01141>

<https://pubs.acs.org/doi/10.1021/acsmchemlett.9b00151>

MIT:

Autonomous discovery, there are some definitions of automation vs autonomous discover but the references here would be most useful.

<https://onlinelibrary.wiley.com/doi/full/10.1002/anie.201909987>

AstraZeneca:

<https://pubs.rsc.org/en/content/articlelanding/2021/RE/D0RE00340A#!divAbstract>

SI:

The HTML of comparisons is really a good addition!

REVIEWER COMMENTS

Authors comments and actions are reported with this color style.

We would like to thank all reviewers for the constructive criticism and for contributing to the overall improvement of the quality of the manuscript.

Reviewer #1 (Remarks to the Author):

The manuscript describe an interesting model for prediction experimental action sequence from a chemical equation. Although the model quality still have much room to improve, this is an important step toward translation of chemical reaction to machine readable/actionable languages. I think the idea is very novel, the manuscript is well written and in general meet the quality for publishing in Nature Communication. I have following comments for the authors to consider:

1. I am wondering if the authors can check if the order of reagents and reactants in the input equation can affect the model prediction.

We performed this check on the 69,352 reactions of the test set (one random molecule shuffling per reaction). It shows the following:

- In 7,922 cases: the placeholders in the actions are updated accordingly.
- In 3,710 cases: the actions are the same, but the placeholders are not updated correctly (i.e., different addition order).
- In 57,720 cases: Different action sequences are suggested.

From these values, it appears that the model fails to recognize that reaction SMILES ordered differently should lead to the same action sequence. It is a sign that the model relies too much on the deterministic order of molecules in the training data, and it shows that the model should be used only with reactions SMILES processed accordingly. As such, the model is similar to an image recognition model that did not learn to recognize objects in images turned upside down. However, while this behavior does not affect the efficacy of the current work, it is possible to improve the performance by using data-augmentation techniques to account for different orders of molecules in identical chemical equations: the training data could be augmented to include reaction SMILES with molecules shuffled randomly.

We added a section in the Supporting Information to present and discuss these results (**Supplementary Note 2**).

2. The model performance according to the reaction class should be discussed. How many reaction classed included in the training set and test set? What is the reaction class overlap among training and test set?

The **Supplementary Note 1** illustrates the class distribution in the training and test sets (among others) and indicates the number of classes missing from those sets.

The performance according to the reaction class is presented in the **Supplementary Note 3**.

3. I am wondering if a transformer pretrained on chemical text can improve the prediction, can the authors comment on that?

The question is very relevant because pretraining can be essential when the actual data to train on is scarce. However, this is not critical for the present work because the training set contains over 500k samples. In our case, the performance is affected mainly by the variability in chemicals, some of which are not sufficiently covered in the 500k samples to learn how changes in chemical structure are reflected in the actions, even for identical chemical reaction classes.

The pure training on chemical text is not likely to improve the predictions mainly because of the minimal overlap between the two vocabularies. In one of our previous works (<https://doi.org/10.1038/s41467-020-17266-6>), we trained transformers on chemical text to extract concise sequence of actions. In that case, the output vocabulary is defined by the words contained in the prosaic text. However, in this work we want the transformer to be focused on action names, placeholders, as well as common reagents (total: 610 tokens) instead of all the vocabulary entries characterizing plain chemical texts.

An interesting point could be the pretraining of the encoder part of the model only, which takes SMILES token as an input, on a larger set of reaction SMILES.

Reviewer #2 (Remarks to the Author):

Vaucher et al. showed that natural language processing techniques can predict the sequence of reaction steps, given reactants and products in SMILES representation. They showed that a trained chemist believes that the machine-predicted synthesis actions are ready for execution in a majority of cases.

I am afraid this paper is significantly below the threshold in terms of significance and impact. As such, I regrettably recommend rejection.

We respect the different opinion of the reviewer. However, predicting an entire sequence of operations automatically is a problem that has no corresponding effort in the existing literature and remains a human task. Here, we demonstrate that this is possible by providing the first machine learning model to do so. In fact, we provide the first ever approach to the problem of compiling a chemical recipe given a set of chemical reactants/reagents with the corresponding desired product. We are sure that the reviewer is able to capture the importance of this task, which has no equivalence in existing literature/technology.

Reaction condition prediction is not a new problem. Walker et al. and Gao et al., cited as ref 5 and 6, already showed that aspects of reaction condition can be predicted. The authors argue that the prediction is not done to the same level of granularity, which is true. However, the authors need to show that this extra level of granularity is accurate, thus meaningful.

The extra level of granularity is essential when considering that reactions are not only about adding reagents and stirring them at a given temperature for a given duration. For the execution of reactions, additional aspects must be considered. As an example, it is important to know whether quenching is needed or not. Or to get rid of unwanted ions or impurities with adequate liquid-liquid extractions (both Extract and Wash actions). There is no doubt that the extra level of granularity is a meaningful component of every chemical reaction.

I think the authors included the expert judgement test to demonstrate chemical accuracy. However, I'm afraid a human chemist would not be able to eyeball the exact sequence of steps. In fact, this is borne out by their data. Their chemist concluded that in 79/500 cases, both the "ground truth" and the predicted sequences are inadequate, despite one of the sequences is in fact from the literature. Therefore, the human test is weak, and insufficient to justify that the extra level of granularity is actually accurate.

The conclusion driven by the reviewer are not reflecting the true nature of the data set. The 79/500 only tells that the ground truth contains a certain level of errors (the 79 examples can be seen in the Supplementary Data 4). In fact, as remarked in the first version of the manuscript ("We expect that the model performance will strongly benefit from quality improvements in training data"), there are important quality issues in the extracted ground truth from patents.

The number of errors in the ground truth (statistically reflecting the original data set) confirms that the human test is indeed a good idea. First, it shows that in the future, additional efforts will be needed to improve the quality of this data. Second, it proves that to a human's eye, the machine-learning model makes predictions that are equally adequate as the ground truth (see other statistics beyond the 79/500 reported by the reviewer) – and therefore, that the model presented in this work will benefit from improvements in the ground truth data.

I would suggest that the meaningful test is (perhaps unfortunately) prospectively doing the experiments. This was the approach taken by several recent papers, e.g. Coley et al. (ref 7), and in the context of retrosynthesis, Mikulak-Klucznik et al. (<https://doi.org/10.1038/s41586-020-2855-y>).

We disagree on this statement. To be statistically meaningful, the validation of trained data-driven models must be done using previously collected data. Performing new experiments has sense only to test models on different data distributions, provided the experiments really explore different part of the chemical phase space. To validate the model, there is no difference in running real experiments (like in the reference provided by the reviewer) or using existing data, if the two belong to the same distribution.

Moreover, it is interesting to observe that the algorithms underlying the works mentioned by the referee were presented in earlier papers and validated using collected data only, without experiments. This seems to contradict the reviewer's statement.

If the authors are planning to do experiments, I suggest:

(a) Trying test set reactions where the model predicts a different sequence of steps compared to literature, and

(b) Trying reactions on unseen substrates, e.g. 4-5 mechanistically different but common reaction classes, and 4-5 reactions per class, making sure that the substrates are chemically very different from the any substrates in the dataset. (b) is important because both the training and testing reactions are "successful" reactions (at least good enough to be mentioned in a patent). This is a significant bias, because reactions often fail!

We thank the referee for these suggestions. We will consider them in a future publication.

Reviewer #3 (Remarks to the Author):

The authors have designed a method that can create recipes for executing chemical synthesis. The paper is well written and certainly publishable in a more technical journal, however, it seems to lack the novelty needed to be publishable in Nature Communications.

Arguably the most interesting part, the text based extraction of the reaction data, has already been published in Nature Communications, reference [9] in this manuscript.

The chemistry community seems to agree that **reaction protocol prediction** is a missing step for the end-to-end automated synthesis of molecules. See, for instance, Fig. 1 of DOI: 10.1126/science.aax1566, <https://science.sciencemag.org/content/sci/365/6453/eaax1566/F2.large.jpg>, where the only step with no existing approaches is the one presented in this manuscript.

We present here a model that predicts the detailed sequence of actions to execute a chemical reaction given as input the reactants/reagents with the corresponding desired product. This has never been addressed in existing work.

For these reasons, we think that our work is novel enough for Nature Communications.

It is truly remarkable that the the recent article from the Cronin lab is not cited <https://science.sciencemag.org/content/370/6512/101/tab-article-info>

The article has very similar goal as this one and in addition includes experimental validation. The article must be known to the authors since one of them has been engaged in a discussion with Professor Cronin on twitter.

We are surprised by this comment: the link between the work of Mehr et al. and the present manuscript is non-existent.

Mehr et al. is about the extraction of experimental steps from the literature using a rule-based system and the execution of these extracted steps on a robotic system.

The present manuscript, instead, aims to predict experimental steps for arbitrary chemical equations. The Mehr et al. paper is linked a lot more closely to our previous work (<https://doi.org/10.1038/s41467-020-17266-6>) than to the current one.

However, we now include the reference to that work as an example (among many others) of today's attempts at automation of chemical reactions on robotic hardware.

Reviewer #4 (Remarks to the Author):

The manuscript titled “Inferring experimental procedures from text-based representations of chemical reactions” builds on previous work by the same authors by extending extracted chemical procedures of exact molecules to a machine learning model that is able to predict procedure operations for unreported molecules. They compare their transformer model to a baseline that mimics how chemists generally apply known procedures to new compounds using a nearest neighbor search based on chemical structure. The analysis for chemical operations can be very difficult since in many cases, there are a multitude of different procedures in various different orders that may be successful for the desired chemical transformation. The authors do a thorough analysis despite the difficulty in analysis.

The article is suggested to be published after addressing the following concerns.

We thank the referee for the detailed review of our manuscript and constructive comments below.

Overall:

The publications on experimental procedures tend to be getting further and further away from being accessible or even usable for the chemistry community. The following make it difficult to implement and validate results:

1) The procedures sequences seem to be tailored for an IBM type automated system. More specifically, only batch chemistry with an ability to program a robot to do these specific actions. This can be acknowledged in the introduction when the authors state “the programming of the robotic systems remains...”. This is important because end users need to know the domain of applicable systems where the predictions can be leveraged. For example, these actions could not be used for flow, parallel chemistry, or HTE in the current form (one could envision how to extend to other systems, but not possible with these action sequences).

It is true that the data used for training the model is limited to batch organic chemistry. We clarified this in the updated manuscript.

The action types and associated properties are not limited to IBM systems. In fact, this work does not even introduce the specification of automation hardware for chemical synthesis because the aim is to be of wide breadth and applicability in different automation contexts. Nothing prevents the current work from being used even in traditional lab, where a chemist can use the generated recipe for a manual execution of all the predicted steps.

The actions were defined in our previous work (<https://doi.org/10.1038/s41467-020-17266-6>) as general "chemist" actions representing the most common operations applied in batch organic chemistry. For execution on an automated system (which we will present in a future work), these "chemist" actions are then converted to instructions specific to the automated system.

The approach and algorithms are not limited to batch organic chemistry – but the bottleneck for application to other domains is the lack of adequate experimental procedure data.

2) Use of closed source data. The reviewer understands that there are not other high quality sources of procedure data available but

We understand the limitation incurred by the use of closed source data and share the concern of the referee that no other high-quality source of procedure data exists (yet).

We updated the manuscript to indicate, in the Data Availability section, that the data set we generated is available from us upon request and upon proof of a valid Pistachio license.

The referee's comment is not complete; we are not sure what the referee meant to say after "but". We will be happy to consider an updated comment for our revised paper.

3) Use of NextMove software to resolve chemical names. While seems robust this is just another closed source package a user would need to have access to in order to repeat this work with their own data. Are there discussions with NextMove to open source the mappings? If not, this could be changed to using the open source PubChem API for resolving synonyms. Eg. For iPrOH <https://pubchem.ncbi.nlm.nih.gov/rest/pug/compound/name/iPrOH/synonyms/JSON> then you can resolve smiles by using the CID <https://pubchem.ncbi.nlm.nih.gov/rest/pug/compound/CID/3776/property/CanonicalSmiles/JSON> where you can replace the number after CID/ to get the smiles for the compounds. This at least would give the ability for others to use the software, even if they can't use the same dataset.

It is true that the current name-to-SMILES and SMILES-to-name services were generated from Pistachio data. We did not contact NextMove about this because we expect they have their own conversion service for these tasks. In our case, generating those mappings from the Pistachio data was the most practical way of enabling efficient conversions between names and SMILES strings, but other approaches are likely to be equally adequate.

In fact, these services are implemented as classes deriving from interfaces (as abstract Python classes) called NameToSmiles and SmilesToName. As such, the specific implementation (Pistachio-based or PubChem-based) is easily interchangeable.

An important consideration for the generation of the dataset is the number of calls to such functions. Considering the number of reactions to process and an average of ~10 calls to name-to-SMILES and SMILES-to-name each, the number of requests to the PubChem servers may be prohibitively large.

The repository that we open-sourced contains the NameToSmiles and SmilesToName interfaces (<https://git.io/Jt1c5> and <https://git.io/Jt1cF>). We also provide an implementation based on Python dictionaries for these interfaces. They are used in two of the examples of the GitHub repository.

- From the chemist's viewpoint, accuracy for procedures is very important. Depending on the reaction type, safety is a really big issue. It is understood that the ML models should learn compatibilities such as a reagent that decomposes at high temperature would never be recorded so the model should not propose it. However, from a risk assessment view in a chemistry department or in industry, there needs to be VERY compelling evidence that these incompatibilities are never executed even robotically where an explosion or fire could hurt human operators and damage equipment/buildings.

We agree that the action sequences suggested by any model should not be executed directly in a laboratory unless there is a strong conviction that the model is absolutely safe.

In this context, one can envision a first level of validation with hard-coded rules (or another ML model) for safety checks, and even human validation of the synthesis actions before the actual execution.

In the Discussion section, we now explicitly state that the predicted actions should not be executed without proper review.

Introduction:

- “The reasoning behind this approach is that the execution of unknown chemical reactions should...”. In this sentence “unknown chemical reactions” should be changed. While the reviewer understands what is being stated, a chemist views this as an undiscovered reaction (ie new reaction type or class) when the authors are really trying to say novel reactant combinations but still using known reaction types.

We updated the sentence to avoid this confusion.

- “Hence, in practice, despite the considerable benefits provided by current AI algorithms...”. Most readers will disagree since there has not been a large demonstrated impact (yet) of AI on synthesis planning and execution. Mostly what we see is vignettes of one-off cases where AI helped. Citations would be needed to backup this claim OR change the sentence to “despite the possible benefits provided by current AI algorithms”.

We updated this sentence.

- The comparison of ML procedures to machine programming is not complete. The similarities are there but it is not clear what the authors are trying to state with this comparison. If it is just to acknowledge that they are building on previous knowledge in other fields, then that is understandable. However, chemists might not see the connection and the major difference in the two predictions is that one misprediction in chemistry can result in major safety issues where in programming there is lower risk associated with poor predictions.

With this paragraph, we simply would like to comment on the fact that our model can be considered to write code for chemical synthesis, which we think is an interesting way to view our approach. We understand that this intention was not very clear in the initial manuscript. We reformulated this paragraph and moved it to later in the introduction.

- “The models do not contain any human encoded chemical knowledge...” While this is true, everything still relies on human coded SMILES (which is a syntax the model is learning), human coded extraction of data etc... This is not to say that this should necessarily be changed but to alert the authors that there are other interpretations.

We were referring to the actual machine learning models, but it is true that this can be misunderstood. We deleted that part of the sentence.

- Distribution of reactions that the expert chemist looks at.

We interpret this comment as the desire to see how the 500 reactions assessed by the expert chemist are distributed. This is now shown in the **Supplementary Note 4**.

Results:

- How is the “commonly used reagents” determined? Frequency? If so, what is the threshold?

This is correct. A list of reagents and associated counts was generated from all the compound names that could not be associated with molecules in the reaction equation from the corresponding procedure. Then, all compounds with a count of 60 or more (total: 190 compounds) were manually reviewed, removing compound names corresponding to

- mixtures of compounds;
- unspecific compounds ("product", "compound 52", "salt", etc.);
- OCR errors;
- Reagents likely playing an essential role in the reaction (i.e., missing from the chemical equations).

We added this description in the Supporting Information, as the **Supplementary Note 5**.

- There is a lot of discussion in the throughout the whole manuscript about the inaccuracy of “overnight”. To a chemist this actually gives a wealth of information, even if it is not useful for automated extraction of procedures. Information gathered from overnight could be that duration is probably between 8-16 hours, an 8 hour range which is not that different of a range than the authors @3@ token (7 hours) and is more specific than the @4@ token. Chemists also gather that the reaction is very likely safe for extended periods of time without supervision and the reactants and products do not decompose over that time. Despite the limitations of overnight in text, the authors defined ranges do not seem to be much more specific and hence the long discussion unwarranted. I think the authors are just trying to state the difficulty in extraction since procedures are very heterogeneous and not standardized.

We reformulated the discussion of overnight and of the specification of temperatures and durations to better state that the exact values are not essential, but rather that the values should be within adequate ranges.

- “These two modifications simplify the design and improve the performance of all computational models without a significant impact on the quality of the predictions.” Please define what “significant impact” is. How is this measured? Quantitatively or qualitatively?

We clarified this sentence by disentangling the consequence of both types of modifications (compound tokens, duration/temperature tokens), and by deleting "significant impact", which cannot be practically measured for the second type of modification.

- “However, the information extracted from patents lacks a proper coverage across different mass scales to capture the typical patterns of the operational changes when using production quantities compared to laboratory scale” cite relevant literature on dataset problems. There are plenty of perspectives and reviews that also call this out. Or cite relevant literature about dataset comparisons in the chemistry realm.

We included two references addressing this issue of mass scale.

- Are the data splits stratified with respect to the data classes or random? If random, provide the resulting data distributions for train, valid, test.

The data splits are random. We clarified this in the manuscript, and added a plot illustrating the class distribution in the **Supplementary Note 1**.

- Figure 2: does this analysis include reaction class 0.0 unrecognized?

Yes, it does. The frequency of this class changes from 30.7% in the original reaction data to 26.2% in the produced data set. We clarified this in the manuscript.

- It is nice to use the novel fingerprints from ref 13 but it would be advisable to use traditional fingerprints (eg circular/Morgan from RDKit) as well. These would give results that would likely be more similar to how search engines such as Reaxys or SciFinder work which allows for those familiar with this type of nearest neighbor search to better interpret the results.

We recently presented the comparison between traditional fingerprints and our data-driven rxnfp (<https://doi.org/10.1038/s42256-020-00284-w>). There, we show that for classifying reactions from Pistachio, the traditional fingerprints have an accuracy of 41% compared to 82% for the pretrained data-driven rxnfp (note that the present work uses the finetuned fingerprint, which reached over 98% after finetuning). In addition to the big difference in accuracy between both fingerprints, this suggests that a nearest-neighbor approach based on traditional fingerprints will fail to select nearest neighbors from the same reaction class in more than 50% of the cases, which may have problematic consequences when selecting reaction conditions based on this criterion. We therefore believe that traditional fingerprints would not provide much of a baseline for inferring experimental procedures and prefer not to include it in the manuscript.

In addition, one disadvantage of traditional reaction fingerprints have is that they require a reactant-reagent split (as introduced by Schneider et al.) or the knowledge of the reaction center (e.g. ChemAxon). In contrast, the data-driven rxnfp works on raw reaction SMILES without distinguishing between reactants and reagents and knowing the reaction center. To our knowledge, in Reaxys / SciFinder, the reaction center has to be specified. Or alternatively, similarity searches can be done on product or reactants only (ignoring the reagents). How exactly those closed-source tools calculate the similarity is not known and difficult to replicate.

We updated the manuscript to motivate the choice of data-driven reaction fingerprints over the traditional ones.

Model evaluation and comparison:

- “Assuming the prediction of each action is an independent event...” Frankly, this is an erroneous assumption. The whole reason to use a transformer model for sequence translation is because each prediction relies on previous and subsequent predictions, hence the global attention heads. It is the reviewer opinion that this analysis does not add much significance and should be removed which would not impact the manuscript as a whole.

We understand that this paragraph was misleading. We did not intend to claim that the action predictions are independent, but rather to provide a more accessible accuracy value than the full-sequence accuracies: the quality of the predicted action sequences is better than an accuracy of 3.6% would suggest. Although the action predictions are not strictly independent, the analysis assuming that they are provides a number that is easier to comprehend, and, to some extent, is independent of the length of the action sequences.

We reformulated this paragraph to clarify that this is only an assumption made for illustrative purposes. We also moved this paragraph to be positioned after the discussion of the low full-sequence accuracies. In addition, we removed it from the abstract and introduction.

- “However, deep-learning models are preferable because they rely on a learned representation rather than on similarities with data points in the training set.” This is debatable. As stated above from a safety risk assessment point of view, most likely close to 100% of companies would go with nearest neighbor model because there would be less liability.

We understand the reviewer's opinion. The safety point mentioned in a previous comment is likely to apply to both nearest-neighbor and deep-learning models. Depending on the fingerprint used for finding nearest neighbors, it may even be that the nearest neighbor is from a different class (see discussion further above), and executing the reaction following the corresponding experimental procedure may be dangerous. This could potentially be avoided by coupling the prediction to reaction template recognition, but in addition to make the framework less general, the nearest-neighbor model may fail to recognize the presence or absence of some (potentially reactive) functional groups.

The beauty of deep learning models is that they are able to learn patterns depending on multiple aspects of the chemical equation without explicitly specifying what they are (as examples: type of chemical transformation, presence/absence of functional groups, etc.).

We updated the manuscript to motivate this preference.

- Figure 4 add the ground truth to the graph

We added the ground truth to Figure 4. To keep adequate, color-blind-friendly color contrasts, we adapted the color palette there and in Figure 3 as well.

- Chemist analysis. This is a very statistically unsound analysis. A single chemist cannot be the judge of the action sequences. The only result the authors should even claim here is if the sequences are adequate or not. The ranking of which prediction is better should be removed OR the authors should recruit 5 or more chemists, demonstrate the chemists have no ties with their research and thus should be less biased, and get their rankings (and even 5 is probably a pretty low number to draw a conclusion on). Chemists have natural biases on how they run reactions which is clearly evidenced by the multiple recorded sequences for the same reaction in the database.

We would like to clarify that the chemist did not do any ranking. We however realize that we were not fully consistent in the manuscript: we mentioned the option of both sequences being adequate, but then spoke of "preference" or "better sequence". We updated the manuscript to clarify this. In essence, the chemist was only asked, for each of the two given sequences, whether they were adequate or not, which gives a total of 4 cases for each reaction: both are adequate, one of the two is adequate, none of the two is adequate. As a consequence, a "ranking" exists when one sequence is adequate and the other is not. We agree that there should be no other ranking as this would become subjective.

As such, the main information to extract from the chemist analysis is not an exact value of how many predicted sequences are valid or not. What is more relevant is that this analysis showed that on average, the model predictions are as good as the ground truth extracted from the actions. This is a very interesting result, as it suggests that the model is likely to benefit from improvements in the ground truth.

Discussion:

- "For most reactions in a random selection of the data set, the predicted action sequences are considered experimentally adequate (313 out of 500) by an expert chemist in a blind assessment. The remaining 187 predictions out of 500 are considered inadequate. The same assessment reveals the presence of 201 inadequate procedures in the ground truth" If the authors are a claim these 500 predictions are representative, then the natural thought to the reader is that 201 out of 500 for the ground truth should be representative as well. This is

40% of the dataset that is not fit for use. This calls into question the methods that are used to extract the sequences. The authors should comment about this in greater detail.

The reviewer's analysis is correct: much of what we call the ground truth is inadequate. We discussed this already in the initial manuscript but have now commented it in greater detail.

To learn from this and improve future versions of the data set and model, we already obtained some insight by inspection of the HTML file given in the supporting information.

For instance, a part of the sequences marked as incorrect is due to incorrect/incomplete reaction schemes (examples: 28, 66, 109, 117, 124, 136, ...). These are errors/misses in the chemical structures (SMILES) of the underlying patent data. Potentially, some filters could remove such reactions from the data set to train (and test) on.

Also, as mentioned in the initial manuscript, we are aware of inadequate data points for reactions involving hydrogenation (examples: 30, 40, 69, 92, ...). This will be improved by allowing for hydrogen as atmosphere during construction of the data set.

In addition to this, the model for extraction from experimental procedures is being improved (f.i. with additional hand annotations) and made more robust. This will also result in larger future dataset.

Seeing that the model is performing on par with the ground truth, we expect that the improvements in the generation of the dataset will result in better-performing models.

Methods:

- "The mapping of extracted names to SMILES strings includes the identification of the base compound name, in this case "sulfuric acid"." Replace the word base because when acid is mentioned one thinks of chemical base. Maybe "root compound name".

We agree and changed it accordingly.

- It would be nice to see how many different ways concentrations, composition ratios etc.. are reported. This would be of great interest to the data scientists and should be fairly trivial to report the number of different ways these are reported, or at least how many the authors consider for standardizing.

In the GitHub repository that we open-sourced, an example script illustrates the simplification of compound names: <https://git.io/Jt1cH>. Concentration or composition ratio values are parsed with the help of regular expressions. In particular, we refer to the classes handling the concentrations, <https://git.io/Jt1cM>, and the composition ratios, <https://git.io/Jt1cS>.

- Multiple sequence entries for the same reaction:

1) What is the criteria for keeping a sequence if the reaction smiles has multiple different entries? Shortest sequence?

When multiple entries exist with the same reaction SMILES, one is picked at random. We updated the manuscript to clarify this.

2) A weight or score could be applied to reactions that have many entries. Could you add a "robustness" score that is trained alongside of the transformer model? Have some range determined by the min and max number of recorded successful sequences for unique reaction

smiles. The output would be sequences and how they would score on a robustness scale. This would be very useful to chemists because it mimics how they would look for procedures to use. For example, they look up an amide coupling and using the same precursors (reagents + reactants) there are 50 reported procedures all that differ. They can infer that it is a robust transformation and will be fairly insensitive to order of addition. You would also be able to use the score to investigate “incorrect” predictions. Hopefully the authors would find that incorrect predictions with a high robustness score would only differ by a trivial order of addition (where safety would be of no concern) or another simple change from the ground truth.

We agree that such a robustness score could be useful. In our opinion, such it should characterize robust combinations of action sequences and types of chemical transformations, rather than only reactions that have many duplicates.

Granting a high robustness score to reactions SMILES with many duplicates with no consideration of the associated action sequences has the risk that they will be marked as robust even if some of the many different ways to perform this reaction are incorrect. As such, it would mark the chemical equation (reaction SMILES) as robust, not the actual action sequence. Providing a robustness value for each reaction of the data set is likely to be a complicated undertaking.

References:

The references are a little sparse. There is a plethora of work in automating chemistry and since that is the end goal of this work, it should be acknowledged. Here are a few to start but there are many references contained in these that should be acknowledged as well.

Lilly automated planning and execution:

<https://pubs.acs.org/doi/full/10.1021/acsmchemlett.8b00488>

<https://pubs.acs.org/doi/full/10.1021/acs.jcim.9b01141>

<https://pubs.acs.org/doi/10.1021/acsmchemlett.9b00151>

MIT:

Autonomous discovery, there are some definitions of automation vs autonomous discover but the references here would be most useful.

<https://onlinelibrary.wiley.com/doi/full/10.1002/anie.201909987>

AstraZeneca:

<https://pubs.rsc.org/en/content/articlelanding/2021/RE/D0RE00340A#!divAbstract>

We included these references, as well as additional ones. We did not include the citation about the virtual assistant, which we think is related to synthesis automation only indirectly.

SI:

The HTML of comparisons is really a good addition!

REVIEWERS' COMMENTS

Reviewer #1 (Remarks to the Author):

All my comments are addressed. I agree that the manuscript can be published in current form.

Reviewer #4 (Remarks to the Author):

The authors have addressed my concerns adequately and it is my recommendation that the manuscript is published.

I only have one suggestion which would not need my re-review in order to publish.

-Is there a way to increase the visibility of some of the filters defined in the github in the methods section? For example Putting the links the authors provided in the response to reviewers in each section where it is discussed, "To obtain the root compound names, we built a series of data-cleaning steps based on heuristics

(https://github.com/rxn4chemistry/smiles2actions/tree/master/smiles2actions/name_filters)...". I find the links provided in the response to reviewers very useful and might not have easily found these in the code. I know this isn't common practice, so I leave it up to the authors to decide if that is useful.

In regard to some of the others reasons for rejection: The previous studies (Gao et al, Walker et al.

etc..) while the first of their kind, are not fully actionable because they only provide a set of reagents, not HOW to execute the reactions. While it is true chemist might be able to infer the procedure, this study by Vaucher et. al. would now allow both human chemists and automated systems to take action on a set of predicted reagents in a more standardized fashion. Additionally, the recommendations for rejection seem to be based on many studies that are only tangentially related to this work at hand. I am very familiar with both the machine learning field and automation, and while there is a link, the work the reviewers cite do not take on the task of extraction/prediction of action sequences but rather implementing action sequences that are human defined and coded. In regard to experimental validation, it is common practice to have data splits for machine learning methods for train/valid/test as the authors did. This is the litmus test of how the data will extrapolate to new molecules because the machine learning model has never seen that data, as a proxy for at the bench validation.

REVIEWER COMMENTS

Authors comments and actions are reported with this color style.

Reviewer #1 (Remarks to the Author):

All my comments are addressed. I agree that the manuscript can be published in current form.

We thank the reviewer again for reviewing our manuscript.

Reviewer #4 (Remarks to the Author):

The authors have addressed my concerns adequately and it is my recommendation that the manuscript is published.

We thank the reviewer again for reviewing our manuscript.

I only have one suggestion which would not need my re-review in order to publish.

-Is there a way to increase the visibility of some of the filters defined in the github in the methods section? For example Putting the links the authors provided in the response to reviewers in each section where it is discussed, “To obtain the root compound names, we built a series of data-cleaning steps based on heuristics (https://github.com/rxn4chemistry/smiles2actions/tree/master/smiles2actions/name_filters)...” . I find the links provided in the response to reviewers very useful and might not have easily found these in the code. I know this isn't common practice, so I leave it up to the authors to decide if that is useful.

This is an interesting suggestion and we agree that it can be helpful in guiding the reader. We added several such links in the Methods section and leave it to the editorial team to keep or remove them.

In regard to some of the others reasons for rejection: The previous studies (Gao et al, Walker et al. etc..) while the first of their kind, are not fully actionable because they only provide a set of reagents, not HOW to execute the reactions. While it is true chemist might be able to infer the procedure, this study by Vaucher et. al. would now allow both human chemists and automated systems to take action on a set of predicted reagents in a more standardized fashion. Additionally, the recommendations for rejection seem to be based on many studies that are only tangentially related to this work at hand. I am very familiar with both the machine learning field and automation, and while there is a link, the work the reviewers cite do not take on the task of extraction/prediction of action sequences but rather implementing action sequences that are human defined and coded. In regard to experimental validation, it is common practice to have data splits for machine learning methods for train/valid/test as the authors did. This is the litmus test of how the data will extrapolate to new molecules because the machine learning model has never seen that data, as a proxy for at the bench validation.

We agree and have nothing to add to this.